# Estimation of Winter Wheat Plant Nitrogen Concentration from UAV Hyperspectral Remote Sensing Combined with Machine Learning Methods

Xiaokai Chen [ID], Fenling Li, Botai Shi and Qingrui Chang *

College of Natural Resources and Environment, Northwest A&F University, Xianyang 712100, China; xiaokaichen@nwafu.edu.cn (X.C.); fenlingli@nwafu.edu.cn (F.L.); 2022060387@nwafu.edu.cn (B.S.)
* Correspondence: changqr@nwsuaf.edu.cn; Tel.: +86-135-7183-5969

**Abstract:** Nitrogen is one of the most important macronutrients and plays an essential role in the growth and development of winter wheat. It is very crucial to diagnose the nitrogen status timely and accurately for applying a precision nitrogen management (PNM) strategy to the guidance of nitrogen fertilizer in the field. The main purpose of this study was to use three different prediction methods to evaluate winter wheat plant nitrogen concentration (PNC) at booting, heading, flowering, filling, and the whole growth stage in the Guanzhong area from unmanned aerial vehicle (UAV) hyperspectral imagery. These methods include (1) the parametric regression method; (2) linear nonparametric regression methods (stepwise multiple linear regression (SMLR) and partial least squares regression (PLSR)); and (3) machine learning methods (random forest regression (RFR), support vector machine regression (SVMR), and extreme learning machine regression (ELMR)). The purpose of this study was also to pay attention to the impact of different growth stages on the accuracy of the model. The results showed that compared with parametric regression and linear nonparametric regression, the machine learning regression method could evidently improve the estimation accuracy of winter wheat PNC, especially using SVMR and RFR, the training set of the model at flowering and filling stage explained 93% and 92% of the PNC variability respectively. The testing set of the model at flowering and filling stages explained 88% and 91% of the PNC variability, the root mean square error of the validation set (RMSE_testing) was 0.82 and 1.23, and the relative prediction deviation (RPD) was 2.58 and 2.40, respectively. Therefore, a conclusion was drawn that it was the best choice to estimate winter wheat PNC at the flowering and filling stage from UAV hyperspectral imagery. Using machine learning methods, SVMR and RFR, respectively, could achieve the most outstanding estimation performance, which could provide a theoretical basis for putting forward the PNM strategy.

**Keywords:** winter wheat; plant nitrogen concentration; unmanned aerial vehicle; machine learning methods; hyperspectral remote sensing

## 1. Introduction

Winter wheat is one of the staple food crops in the world. Nitrogen is one of the most essential macronutrients and plays an essential role in the growth and development of winter wheat. The precision nitrogen management (PNM) strategy is committed to accurately guiding the most reasonable nitrogen fertilizer consumption of crops at an appropriate time and place [1–3]. Therefore, it has become a prerequisite for precision agriculture (PA), which is significant for improving crop yield [4] and quality [5] and reducing environmental pollution [6]. Thus, it is very crucial to diagnose the nitrogen status of winter wheat timely and accurately for applying the PNM strategy to the guidance of nitrogen fertilizer in the field.

Previous evaluations of crop nitrogen status mainly focused on the estimation of leaf nitrogen content (LNC) [7–9]. However, Fageria [10] showed that plant nitrogen concentration (PNC) was widely used as an effective indicator of crop nitrogen diagnosis, which

had a close relationship with crop yield [11,12]. Therefore, researchers tried to estimate crop PNC through different methods, which would be taken as a reference for establishing crop nitrogen nutrition diagnosis methods and set different PNC thresholds according to the differences in crop varieties and growth stages [13–15]. However, the determination of PNC requires destructive sampling in the field and chemical analysis, which is not only expensive but also has a certain lag. The development of remote sensing makes it possible to diagnose crop nitrogen status without damage and at low cost. Researchers tried to monitor the crop nitrogen index using proximal remote sensing sensors such as Dualex 4 (Force-A, Orsay, Paris, France) [16–18], GreenSeeker (Trimble, Westminster, CO, USA) [3,19,20], Multiplex®3 (Dynamax, Elkhart, IN, USA) [21], and RapidSCAN CS-45 (Holland Scientific Inc., Lincoln, NE, USA) [22,23] and achieved encouraging research results. Although the proximal sensing technology has incomparable advantages, such as being free from the interference of background factors such as light and soil [15], it still needs to repeat a lot of measurement work in crop canopy and can only obtain single point information of field crops, which is difficult to achieve regional scale nitrogen nutrition monitoring. At the same time, satellite remote sensing technology has been gradually applied to diagnosing crop nitrogen status in a large area and has yielded a lot [24–26]. Nevertheless, the satellite sensors have fixed transit dates, so it is not very flexible to obtain satellite remote sensing imagery.

Recently, unmanned aerial vehicle (UAV) remote sensing has shown great prospects in PA due to its advantages of small size, flexibility, low cost, and high spatial and temporal resolution, which has been successfully applied to the prediction of leaf area index (LAI) [27], chlorophyll [28], biomass [29], yield [30], etc. For example, Zheng et al. [31] studied the PNC of rice from UAV imagery, and the results showed that narrow-band spectral indices with texture information of UAV might be a promising method for crop growth monitoring. Furthermore, Zha et al. [32] successfully estimated the nitrogen nutrition index (NNI) of rice by combining multi-spectral remote sensing and machine learning regression of fixed-wing UAV; Wang et al. [33] estimated NNI of grass crops based on the multi-spectral camera, and the research results showed the red edge vegetation index was the best among all vegetation indexes. However, previous studies on the assessment of the nitrogen status of these crops mostly focused on a single growth stage. Although the assessment of nitrogen status in the early stage of winter wheat vegetative growth is very important for the guidance of fertilization in the later stage [34,35], the assessment of nitrogen status in the late maturity stage can provide an essential indicator for the yield of crops in the next year [36]. Thus, it is very important to continuously diagnose the nitrogen status in the stages of winter wheat growth and development, which may contribute to the improvement of economic benefits [36,37].

Machine learning regression methods have been widely applied in crop parameter estimation and achieved encouraging research results [35,38–40]. For example, Yang et al. [41] combined the optimized spectral index (OSI) and machine learning methods to estimate the nitrogen status of crops, and the research results showed that the OSI-based RF is a robust and effective model to predict crop PNC at the vegetative growth stage. In addition, the work of Shah et al. [42] revealed that random forest regression (RFR) could reduce the root mean square error (RMSE) better than standard linear regression. However, when using machine learning methods to estimate winter wheat PNC, the influence of different growth stages on model accuracy has not been fully explored.

The main purpose of this study was to use three different prediction methods to evaluate winter wheat PNC at booting, heading, flowering, filling and the whole growth stages in the Guanzhong area based on hyperspectral reflectance data of UAV imaging, including (1) the parametric regression method; (2) linear nonparametric regression methods (stepwise multiple linear regression (SMLR) and partial least squares regression (PLSR)); and (3) machine learning methods (random forest regression (RFR), support vector machine regression (SVMR), and extreme learning machine regression (ELMR)), and pay attention to the impact of different growth stages on the accuracy of the models.

## 2. Materials and Methods

### 2.1. Experimental Design

The experiment was conducted in Qian County (108°07′E, 34°38′N. average altitude: 830 m) in Shaanxi Province, China (Figure 1). A total of 36 plots were set up and involved a common local winter wheat cultivar Xiaoyan 22. The plot area was 9 m × 10 m = 90 m², and 6 levels of nitrogen treatment (0, 60, 120, 180, 240, 300 kg ha$^{-1}$), phosphorus treatment (0, 30, 60, 90, 120, 150 kg ha$^{-1}$) and potassium treatments (0, 30, 60, 90, 120, 150 kg ha$^{-1}$) were set, respectively. Each treatment was repeated twice, and nitrogen, phosphorus and potassium fertilizers were applied in the form of urea, phosphate ($P_2O_5$), and potash chloride ($K_2O$), respectively. Potassium and phosphorus fertilizers were not applied in nitrogen fertilizer treatment. Phosphate fertilizer treatment did not apply potassium fertilizer; nitrogen fertilizer was applied with 1/2 standard nitrogen fertilizer (60 kg ha$^{-1}$); potassium fertilizer treatment did not apply phosphorus fertilizer; and nitrogen fertilizer was applied with 1/2 standard nitrogen fertilizer (60 kg ha$^{-1}$). All fertilizers shall be applied as base fertilizer at one time, and no additional fertilizer will be applied. The management method is the same as that of local conventional winter wheat.

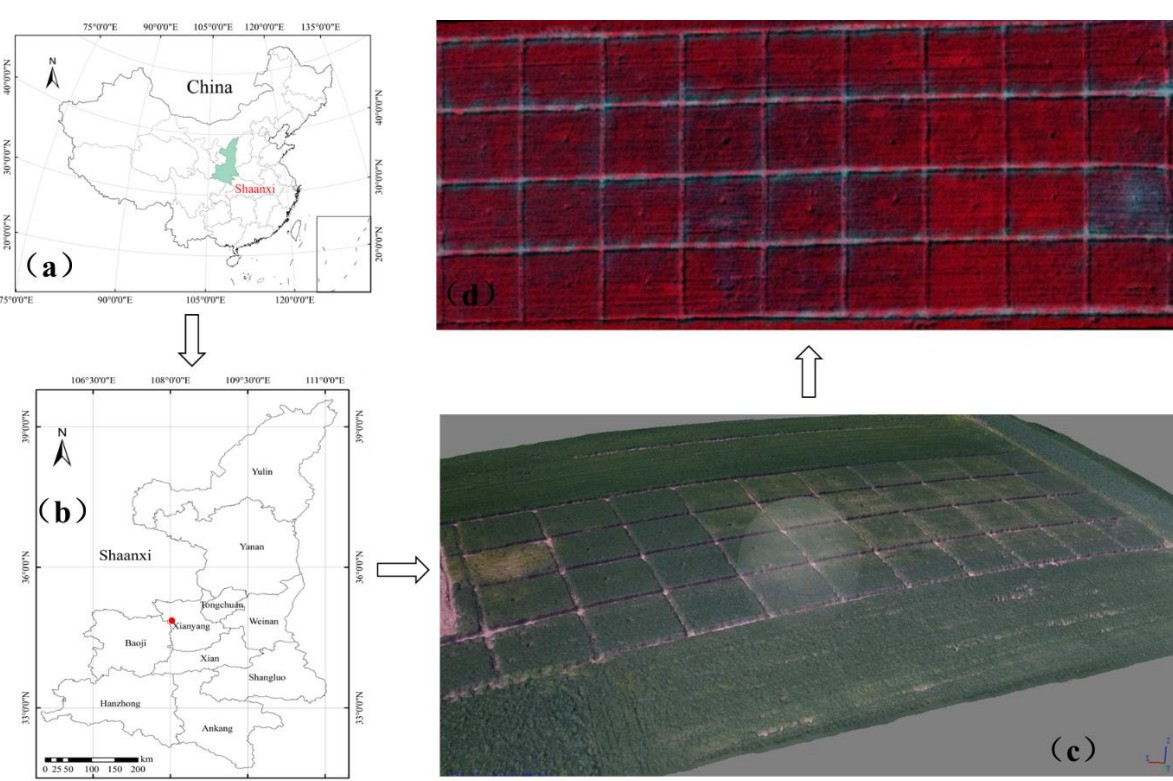

**Figure 1.** (**a**) Geographic location of Shaanxi Province, China; (**b**) geographical location of the study area in Shaanxi Province; (**c**) 3D view of the experimental field; and (**d**) standard false color synthetic display of the experimental field.

### 2.2. Data Acquisition

2.2.1. Acquisition and Processing of UAV Hyperspectral Image

A six-rotor UAV (DJI M600Pro, SZ DJI Technology Co., Shenzhen, Guangzhou, China) as the platform (Figure 2) and a Cubert S185 (S185, Cubert GmbH, Ulm, Germany) hyperspectral imager were used to obtain winter wheat growth stages canopy hyperspectral images in the study area. S185 is a lightweight (470 g), full frame, and real-time imaging spectrometer. It can acquire 125 channels of hyperspectral imagery in the wavelength range of 450 nm to 950 nm. It only takes 1/1000 s to obtain each spectral cube, and the spectral sampling interval is 4 nm. Before acquiring hyperspectral data, calibrate the reference plate

and dark current of the S185 imaging spectrometer, plan the flight route, and set a sampling interval of 1 ms for data acquisition, the flight height at 100 m, and the speed at 6 m s$^{-1}$. The spectrometer lens is vertically downward, and the field angle is 30°. Therefore, 80% of the heading overlap and 70% of the lateral overlap are guaranteed.

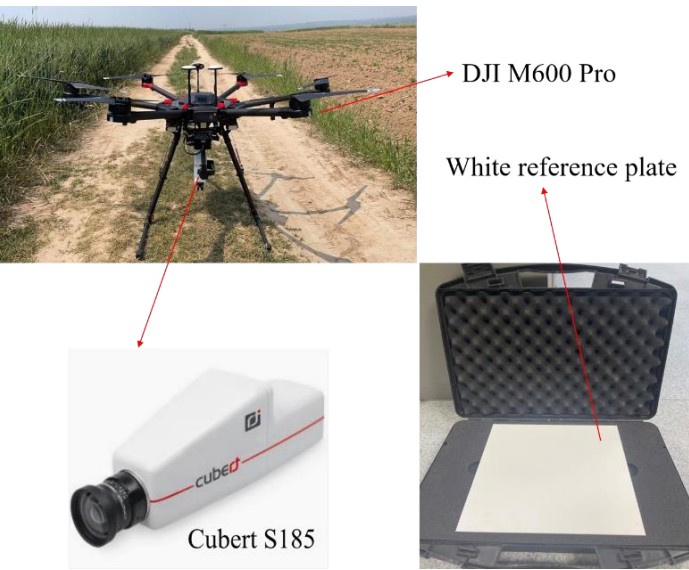

**Figure 2.** Platform (DJ M600 Pro) with UAV sensor, sensor S185, and white reference board with 100% reflectivity.

After acquiring hyperspectral data, the pan sharpens fusion function in the UAV image professional processing software Cubert Utils Touch program is used to fuse hyperspectral and grayscale images. Then, a single gray image is spliced with the aid of the image processing software AgisoftPhotoScan 1.2.4. Next, the Georeferencing function in Arcmap10.6 software combines with the corresponding research area image in Google Earth software to perform geographic registration for the spliced gray image. Finally, according to the geographic coordinate information of the ground sampling points, the Region of Interest (ROI) tool in ENVI 5.3 (The Environment for Visualizing Images) software is used to draw the 30 × 30 cm range limit of the sampling points, then calculate and extract the canopy hyperspectral average reflectance of the target area. The Savitzky–Golay (S–G) smoothing method was used to eliminate the noise information attached to the UAV canopy hyperspectral. UAV imageries were obtained at the booting, heading, flowering, and filling stages of winter wheat.

### 2.2.2. PNC Determination

The aboveground plants of winter wheat with an area of 30 × 30 cm were collected at each plot. Meanwhile, the position information of each sampling point was recorded by Real-time kinematics (RTK). Winter wheat plant samples were placed in an oven at 105 °C for 30 min and dried at 80 °C for about 48 h until the weight of the sample no longer changed. Then the dry sample was crushed, weighed at about 0.2 g, and digested with concentrated $H_2SO_4$ in the presence of a catalyst. The modified Kjeldahl digestion method was used to determine winter wheat PNC.

### 2.3. Analytical Methods

The spectral index (SI) is composed of the reflectivity of specific bands, which can partly eliminate the interference of soil, weather, and other factors, and improve the sensitivity of target parameters. In this study, 10 spectral indexes closely related to crop nitrogen status were used to establish the winter wheat PNC estimation model. The relevant spectral indexes and their calculation formulas are shown in Table 1.

**Table 1.** Spectral index selected in this study and calculation formula.

| Spectral Index | Formula | Reference |
|---|---|---|
| NDVI | $(R_{800} - R_{670})/(R_{800} + R_{670})$ | [43] |
| SAVI | $1.5 * (R_{800} - R_{670})/(R_{800} + R_{670} + 0.5)$ | [44] |
| NRI | $(R_{570} - R_{670})/(R_{570} + R_{670})$ | [45] |
| NDRE | $(R_{790} - R_{720})/(R_{790} + R_{720})$ | [46] |
| OSAVI | $1.16 * (R_{800} - R_{670})/(R_{800} + R_{670} + 0.16)$ | [47] |
| GNDVI | $(R_{750} - R_{550})/(R_{750} + R_{550})$ | [48] |
| mND705 | $(R_{750} - R_{705})/(R_{750} + R_{705} - 2R_{445})$ | [49] |
| CIre | $(R_{750})/(R_{720}) - 1$ | [50] |
| MTCI | $(R_{750} - R_{710})/(R_{710} - R_{680})$ | [51] |
| CIgreen | $(R_{800})/(R_{560}) - 1$ | [50] |

Note: $R_i$ is the reflectance of band i nm. If the reflectivity of S185 cannot correspond to $R_i$ in the calculation formula, replace it with the average of the reflectivity values of the adjacent bands.

First of all, parameter regression was used to explore the relationship between the independent variable and the dependent variable (linear, power function, exponential function, logarithmic function, and unitary quadratic function, etc.) and then establish the PNC estimation model of the whole and single growth stage. Then, stepwise multiple linear regression (SMLR), partial least squares regression (PLSR), random forest regression (RFR), support vector machine regression (SVMR), and extreme learning machine regression (ELMR) models based on SIs were constructed.

As a linear nonparametric regression method, SMLR was often used to evaluate crop nitrogen status [52,53]. Generally, SMLR uses a set of given explanatory variables to subtract or add a variable from all explanatory variables according to specific criteria to determine whether the prediction performance of the model will change [54]. In this study, Akaike's information criterion (AIC) was used as the evaluation standard, and the prediction model was the best when the AIC value was the smallest [55]. The MASS package in the statistical software R was used to accomplish the SMLR modeling and parameter optimization.

PLSR was used to build winter wheat PNC estimation models. PLSR is an innovative bilinear regression approach [56] for assessing multivariate statistical data. For a detailed description of PLS, please refer to our previous work [57]. The plsmod package in the statistical software R was used to accomplish the PLSR modeling and parameter optimization.

RFR was proposed by Breiman [58] in 2001. For a detailed description of RFR, please refer to our previous work [59]. The predicted results were averaged by integrating decision trees after the samples were constantly regressed and sampled several times to generate a training set. The algorithm must primarily optimize two essential parameters: ntree (number of decision trees) and mtry (number of segmentation nodes). In this study, ntree was set to 500, and mtry was set by grid search. The random forest package in the statistical software R was used to accomplish the RFR modeling and parameter optimization.

SVMR is characterized by using kernel, sparse solution, and the Vapnik–Chervonenkis theory to control the number of edges and support vectors [60]. The kernel function with radial basis function (RBF) was used to perform SVMR, and the kernel parameters of RBF kernel σ and regularization parameter C were determined by grid search [39]. The kernlab package in the statistical software R was used to accomplish the SVMR modeling and parameter optimization.

ELMR is an algorithm based on a single hidden layer feedforward neural network. Compared with traditional neural networks, it has the advantages of fast convergence speed and strong generalization ability [61]. The characteristic of ELMR is to initialize the input weights and offsets randomly. By calculating the output weights of hidden layer neurons, the learning speed of the limit learning machine is accelerated. Furthermore, according to the solution method of linear equations, when the sample hidden layer neuron output value matrix is full rank, only the one-time operation of matrix inversion is required to obtain the hidden layer neuron weight. Based on this, the application of the ELMR model usually requires multiple runs and a prediction model with high recording accuracy [62].

The elmNNRcpp package in the statistical software R was used to accomplish the ELMR modeling and parameter optimization.

In this paper, the data sets of the whole and single growth stages were divided into a 3:1 ratio to construct training and testing set by the caret package in the statistical software R. The coefficient of determination ($R^2$), root mean square error (RMSE), and relative prediction deviation (RPD) were used to evaluate the accuracy of the winter wheat PNC estimation models. RPD is the ratio between the standard deviation (SD) of the test set and RMSE [63]. RPD < 1.40 represents that the model does not have prediction performance; 1.40 < RPD < 2.00 represents the rough prediction ability of the model; RPD > 2.00 represents the excellent prediction ability of the model. In the construction process of winter wheat PNC estimation models, grid search is used to determine the super parameters, and 10-fold cross-validation repeated 5 times is used to identify the optimal relevant parameters of each model on the calibration set.

## 3. Results

### 3.1. Statistical Description of Winter Wheat PNC

The statistical description of the winter wheat whole and single-growth stage PNC is shown in Table 2. A total of 144 samples were obtained in the whole growth stage. The variation range of PNC is 5.39~34.69, with a mean of 12.41, an SD of 4.80, and a CV of 38.65%. Samples were obtained from the booting stage, with a max of 34.69, a min of 11.28, a mean of 17.91, an SD of 4.97, and a CV of 27.74%. The PNC ranges from 7.82 to 18.96, with a mean of 12.49, an SD of 2.88, and a CV of 23.03% during the heading stage, with a max of 15.52, a min of 6.27, a mean of 10.20, an SD of 2.55, and a CV of 25.01% during the flowering stage. The PNC varies from 5.39 to 14.11, with a mean of 9.04, an SD of 2.59, and a CV of 28.69% during the filling stage. The mean value of PNC decreased with the advance of the winter wheat growth stage because of the dilution effect [5].

**Table 2.** Statistical description of winter wheat PNC (g kg$^{-1}$) for whole and single growth stage.

| Growth Stages | The Number of Samples | Max | Min | Mean | SD | CV (%) |
|---|---|---|---|---|---|---|
| Whole | 144 | 34.69 | 5.39 | 12.41 | 4.80 | 38.65 |
| Booting | 36 | 34.69 | 11.28 | 17.91 | 4.97 | 27.74 |
| Heading | 36 | 18.96 | 7.82 | 12.49 | 2.88 | 23.03 |
| Flowering | 36 | 15.52 | 6.27 | 10.20 | 2.55 | 25.01 |
| Filling | 36 | 14.11 | 5.39 | 9.04 | 2.59 | 28.69 |

### 3.2. Performance of Prediction Models for PNC Constructed by Parameter Regression

The correlation heat diagram between the spectral index and PNC is shown in Figure 3. Table 3 lists the performance of the prediction model for PNC constructed by two spectral indexes with the best fitting degree in the whole and single growth stages. What needs our attention is that the PNC prediction models based on the spectral index are all quadratic models, and the spectral indexes selected in each growth period are MTCI and CIre. For the whole growth stage, the PNC estimation model based on MTCI has the lowest RMSE$_{cv}$, and its accuracy is slightly higher than CIre. The best performance parameter regression model for a single stage can explain 47%, 83%, 86%, and 77% of PNC variability in the booting, heading, flowering, and filling stages. The performance of the PNC prediction model is ordered as follows: flowering > filling > heading > whole > boosting. The distribution of observed and predicted PNC (g kg$^{-1}$) of prediction models in Table 3 are shown in Figure 4.

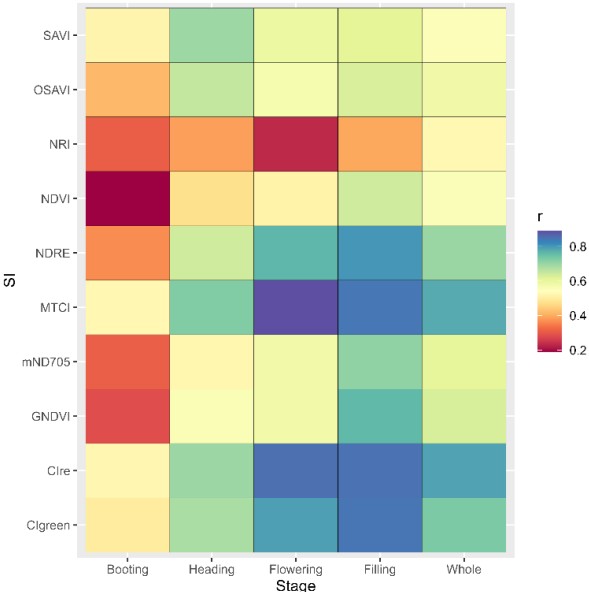

**Figure 3.** The correlation heat diagram between the spectral index and PNC.

**Table 3.** Performance of prediction model for PNC constructed by two spectral indexes with the best fitting degree in the whole and single growth stage.

| Growth Stages | Spectral Indices | Model | Calibration Set (Cross-Validated) | | Testing Set | | |
|---|---|---|---|---|---|---|---|
| | | | $R^2_{cv}$ | $RMSE_{cv}$ | $R^2_{testing}$ | $RMSE_{testing}$ | RPD |
| Whole | CIre | Q | 0.70 | 2.73 | 0.64 | 2.71 | 1.56 |
| | MTCI | Q | 0.71 | 2.65 | 0.64 | 2.67 | 1.59 |
| Booting | CIre | Q | 0.49 | 3.60 | 0.15 | 3.94 | 1.08 |
| | MTCI | Q | 0.47 | 3.70 | 0.47 | 3.30 | 1.29 |
| Heading | MTCI | Q | 0.57 | 1.89 | 0.83 | 1.56 | 1.71 |
| | CIre | Q | 0.52 | 2.00 | 0.63 | 1.90 | 1.40 |
| Flowering | MTCI | Q | 0.90 | 0.84 | 0.86 | 0.97 | 2.19 |
| | CIre | Q | 0.78 | 1.22 | 0.80 | 1.07 | 1.98 |
| Filling | CIre | Q | 0.75 | 1.22 | 0.77 | 1.53 | 1.92 |
| | MTCI | Q | 0.74 | 1.24 | 0.75 | 1.55 | 1.91 |

Note: Q stands for Quadratic model.

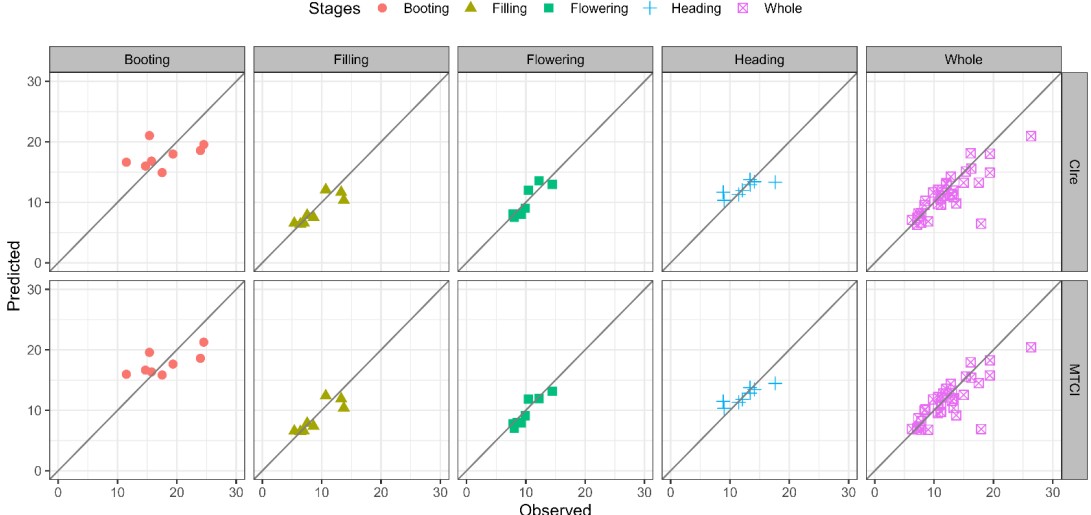

**Figure 4.** Distribution of observed and predicted PNC (g kg$^{-1}$) of prediction models in Table 3. The 1:1 line is depicted with gray lines.

### 3.3. Performance of Prediction Models for PNC Constructed by Linear Nonparametric Regressions

Table 4 lists the PNC prediction model equations formed by SIs screened by SMLR in the whole and single growth stages when the AIC value is the minimum. It can be observed that the number of selected SI varies from four to seven. The performance of prediction models for PNC constructed by SMLR and PLSR in the whole and single growth stages are given in Table 5. The best prediction models for estimating PNC based on linear nonparametric regression methods at booting, heading, flowering, filling, and the whole growth stage of winter wheat are SMLR ($R^2_{testing}$ = 0.64, $RMSE_{testing}$ = 2.69, RPD = 1.58), PLSR ($R^2_{testing}$ = 0.22, $RMSE_{testing}$ = 3.92, RPD = 1.09), PLSR ($R^2_{testing}$ = 0.60, $RMSE_{testing}$ = 1.96, RPD = 1.36), SMLR ($R^2_{testing}$ = 0.83, $RMSE_{testing}$ = 1.17, RPD = 1.80), and PLSR ($R^2_{testing}$ = 0.80, $RMSE_{testing}$ = 1.50, RPD = 1.97). Figure 5 shows the distribution of observed and predicted PNC (g kg$^{-1}$) of prediction models in Table 5.

**Table 4.** Selection of SI in the SMLR model and AIC values.

| Growth Stages | Formula | AIC |
|---|---|---|
| Whole | $-22.384 + 22.572 * SAVI - 2.413 * CIgreen + 15.364 * CIre - 115.188 * NDRE - 45.186 * mND705 + 2.270 * MTCI + 110.095 * GNDVI$ | 201.15 |
| Booting | $36.348 + SAVI * 580.431 - 502.181 * OSAVI - 3.246 * CIgreen + 7.213 * MTCI + 67.614 * NRI$ | 63.14 |
| Heading | $-13.893 + 239.041 * NDVI + 562.864 * SAVI - 670.480 * OSAVI - 61.127 * mND705 + 4.501 * MTCI - 33.247 * NRI$ | 33.41 |
| Flowering | $13.688 - 574.442 * NDVI - 1375.346 * SAVI + 1749.35 * OSAVI + 3.005 * MTCI$ | −5.67 |
| Filling | $11.226 - 144.715 * SAVI + 173.963 * OSAVI + 14.772 * CIre - 3.737 * MTCI - 49.850 * NRI - 67.265 * GNDVI$ | 9.66 |

**Table 5.** Performance of prediction model for PNC constructed by SMLR and PLSR in the whole and single growth stage.

| Growth Stages | Methods | Calibration Set (Cross-Validated) | | Testing Set | | |
|---|---|---|---|---|---|---|
| | | $R^2_{cv}$ | $RMSE_{cv}$ | $R^2_{testing}$ | $RMSE_{testing}$ | RPD |
| Whole | | 0.77 | 2.45 | 0.64 | 2.69 | 1.58 |
| Booting | | 0.76 | 2.81 | 0.43 | 4.29 | 0.99 |
| Heading | SMLR | 0.76 | 1.63 | 0.13 | 2.64 | 1.01 |
| Flowering | | 0.92 | 0.83 | 0.83 | 1.17 | 1.80 |
| Filling | | 0.86 | 1.01 | 0.73 | 1.87 | 1.57 |
| Whole | | 0.76 | 2.43 | 0.61 | 2.87 | 1.47 |
| Booting | | 0.28 | 4.30 | 0.22 | 3.92 | 1.09 |
| Heading | PLSR | 0.48 | 2.07 | 0.60 | 1.96 | 1.36 |
| Flowering | | 0.90 | 0.82 | 0.85 | 1.19 | 1.79 |
| Filling | | 0.70 | 1.34 | 0.80 | 1.50 | 1.97 |

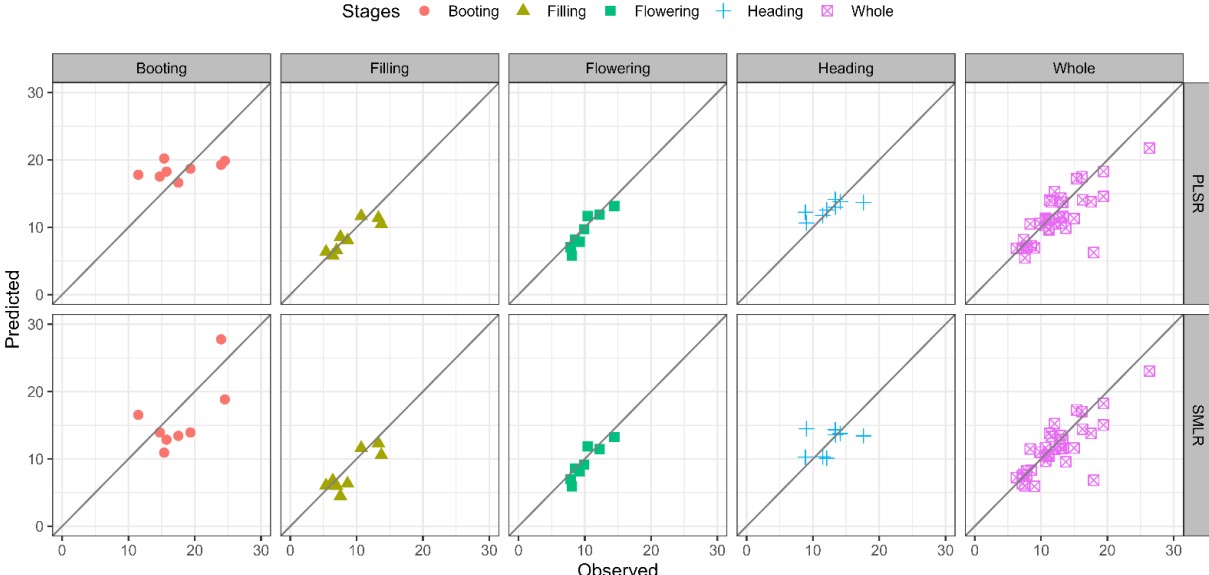

**Figure 5.** Distribution of observed and predicted PNC (g kg$^{-1}$) of prediction models in Table 5. The 1:1 line is depicted with gray lines.

### 3.4. Performance of Prediction Models for PNC Constructed by Machine Learning Methods

Table 6 lists the performance of the prediction model for PNC constructed by RFR, SVMR and ELMR in the whole and single growth stages. In general, the best prediction models for estimating PNC based on machine learning methods at booting, heading, flowering, filling and the whole growth stage of winter wheat are SVMR ($R^2_{testing}$ = 0.40, RMSE$_{testing}$ = 3.44, RPD = 1.24), ELMR ($R^2_{testing}$ = 0.91, RMSE$_{testing}$ = 1.42, RPD = 1.87), SVMR ($R^2_{testing}$ = 0.88, RMSE$_{testing}$ = 0.82, RPD = 2.58), RFR ($R^2_{testing}$ = 0.91, RMSE$_{testing}$ = 1.23, RPD = 2.40), RFR ($R^2_{testing}$ = 0.69, RMSE$_{testing}$ = 2.51, RPD = 1.69). Figure 6 shows the distribution of observed and predicted PNC (g kg$^{-1}$) of prediction models in Table 6.

**Table 6.** Performance of prediction model for PNC constructed by RFR, SVMR and ELMR in the whole and single growth stage.

| Growth Stages | Methods | Calibration Set (Cross-Validated) | | Test Set | | |
|---|---|---|---|---|---|---|
| | | $R^2_{cv}$ | RMSE$_{cv}$ | $R^2_{test}$ | RMSE$_{test}$ | RPD |
| Whole | | 0.94 | 1.28 | 0.69 | 2.51 | 1.69 |
| Booting | | 0.88 | 1.74 | 0.36 | 3.45 | 1.24 |
| Heading | RFR | 0.89 | 1.06 | 0.72 | 1.48 | 1.8 |
| Flowering | | 0.95 | 0.6 | 0.76 | 1.1 | 1.8 |
| Filling | | 0.92 | 0.73 | 0.91 | 1.23 | 2.4 |
| Whole | | 0.89 | 1.68 | 0.64 | 2.83 | 1.5 |
| Booting | | 0.93 | 1.61 | 0.4 | 3.44 | 1.24 |
| Heading | SVMR | 0.51 | 2.16 | 0.72 | 1.9 | 1.41 |
| Flowering | | 0.93 | 0.69 | 0.88 | 0.82 | 2.58 |
| Filling | | 0.85 | 0.96 | 0.71 | 1.8 | 1.63 |
| Whole | | 0.75 | 2.47 | 0.65 | 2.68 | 1.58 |
| Booting | | 0.27 | 4.22 | 0.02 | 11.3 | 0.44 |
| Heading | ELMR | 0.53 | 1.99 | 0.91 | 1.42 | 1.87 |
| Flowering | | 0.87 | 0.88 | 0.84 | 1.1 | 1.93 |
| Filling | | 0.66 | 1.47 | 0.76 | 1.56 | 1.91 |

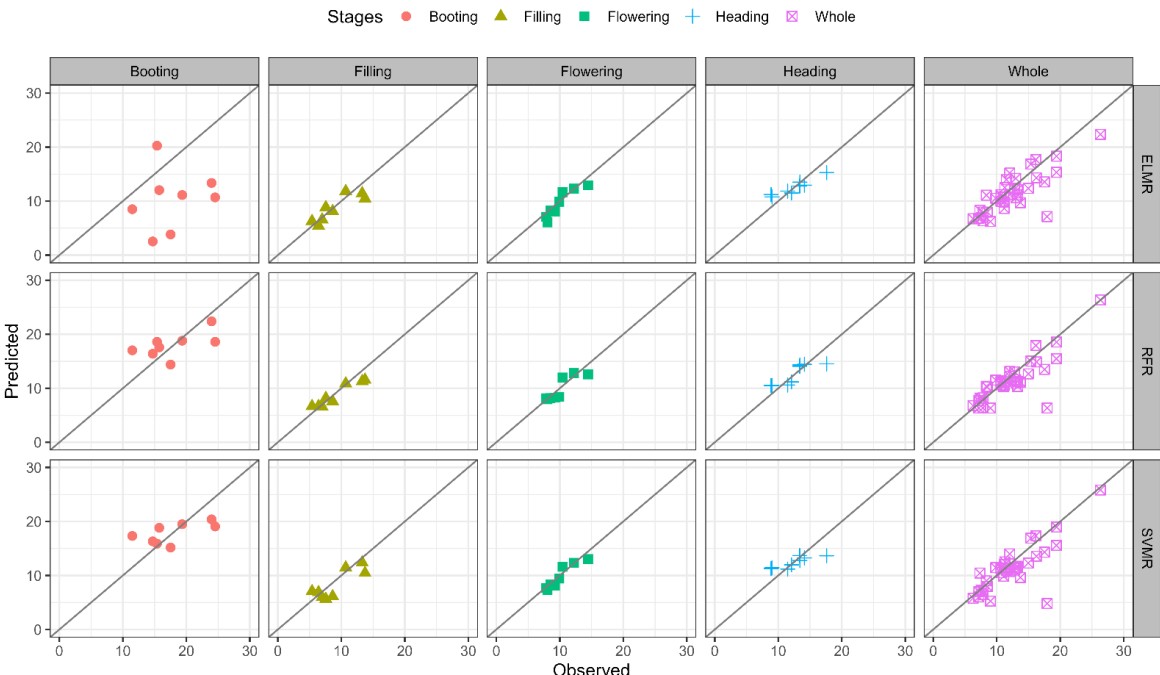

**Figure 6.** Distribution of observed and predicted PNC (g kg$^{-1}$) of prediction models in Table 6. 1:1 line is depicted with gray lines.

### 3.5. Model Accuracy Comparison

As shown in Figure 7, the RPD was used further to evaluate the model accuracy of winter wheat PNC based on the whole growth stage and boosting, heading, flowering, and filling stage. In the winter wheat boosting stage, it is obvious that the winter wheat PNC estimation model based on univariate regression, linear nonparametric regression, or machine learning regression does not have the prediction ability (PRD < 1.40) and cannot be used to estimate the distribution of PNC. Although the estimation model based on spectral index MTCI has the highest accuracy (RPD = 1.29), it is superior to linear nonparametric regression (SMLR, PLSR) models and machine learning regression (RFR, SVMR, ELMR) models. During the heading stage, three PNC estimation models have outstanding performance in the seven estimation models constructed and can be used to estimate PNC (1.40 < RPD < 2.00). Among them, the ELMR model has the highest accuracy in predicting PNC (RPD = 1.87), followed by the RFR model (RPD = 1.80), and finally, a univariate regression model based on spectral index MTCI (RPD = 1.71). The SMLR model and PLSR model perform the worst, and their prediction accuracy is lower than the parameter regression PNC estimation model based on spectral index CIre. During the winter wheat flowering stage, it is gratifying to see that all PNC estimation models have significantly improved. Although the PLSR model performs the worst (RPD = 1.79), the RPD values of all PNC prediction models exceed 1.40. The estimation model based on spectral index MTCI and machine learning SVMR has strong PNC prediction ability. The RPD value of the SVMR model is as high as 2.58, and the RPD of the regression model based on the spectral index MTCI is 2.19. It is worth noting that the prediction model following the SVMR model is a parameter regression model based on the spectral index MTCI, followed by EMLR (RPD = 1.93), SMLR (RPD = 1.80), RFR (RPD = 1.80), and PLSR (RPD = 1.79). At the filling stage, all models have good estimation ability (RPD > 1.40). The PNC estimation model based on RFR has the best prediction performance (RPD = 2.40). The PNC estimation model based on SMLR has the worst prediction performance (RPD = 1.57). Rank the RPD values of the prediction models as RFR (RPD = 2.40) > PLSR (RPD = 1.97) > CIre (RPD = 1.92) > MTCI (RPD = 1.91), EMLR (RPD = 1.91) > SVMR (RPD = 1.63) > SMLR (RPD = 1.57). In the whole growth stage, it seems that the distribution of RPD values is relatively concentrated, which indicates that the prediction accuracy of all PNC estimation models is similar and

can be used to estimate the distribution of PNC (RPD > 1.40). A conclusion can be drawn that the prediction model based on RFR is also the best (RPD = 1.69). When predicting PNC, the PLSR model has the lowest prediction accuracy (RPD = 1.47), which is lower than the parameter regression PNC estimation models based on spectral index CIre and MTCI. In general, the best prediction models for estimating PNC at booting, heading, flowering, filling, and the whole growth stage of winter wheat are MTCI (RPD = 1.29), ELMR (RPD = 1.87), SVMR (RPD = 2.58), RFR (RPD = 2.40), RFR (RPD = 1.69).

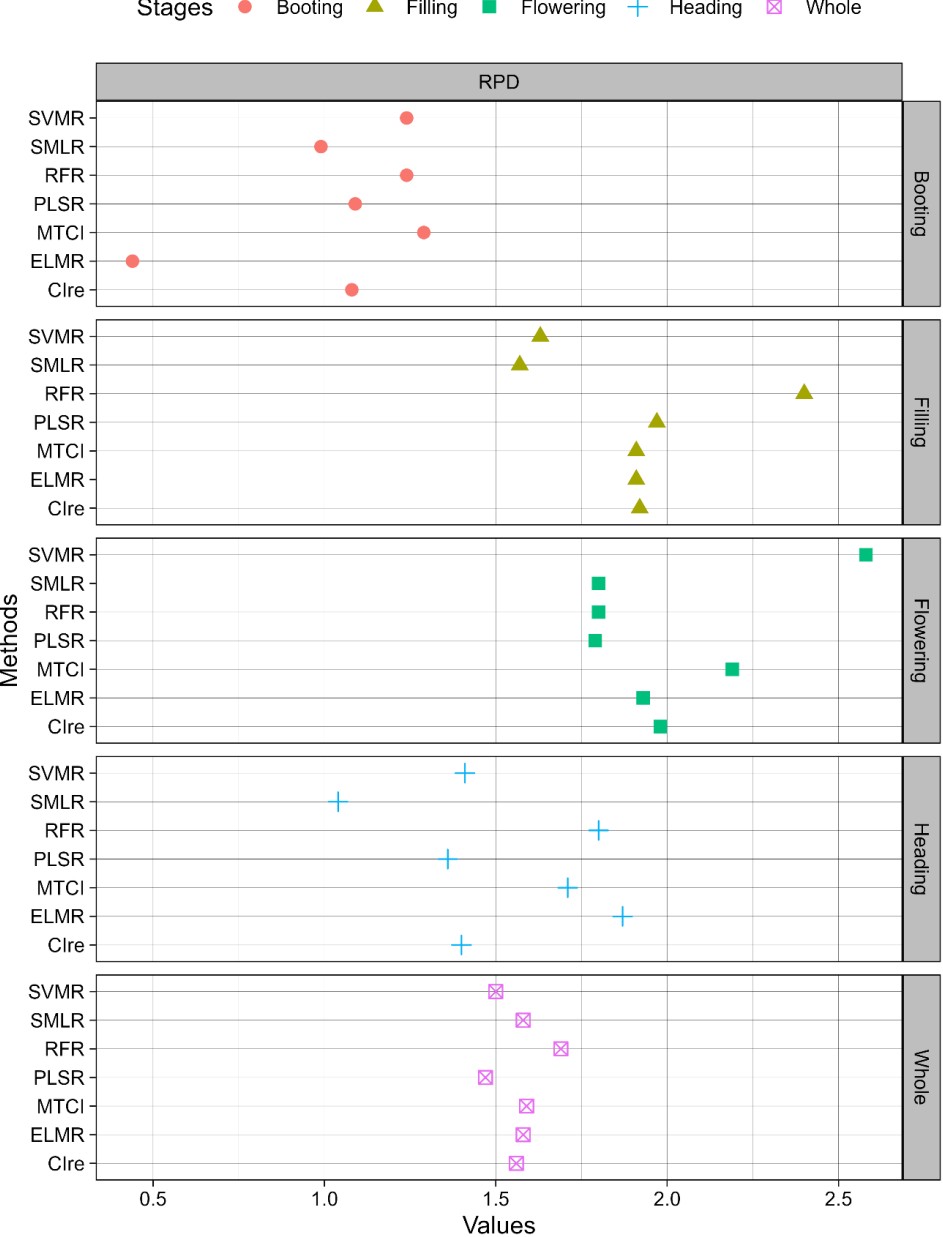

**Figure 7.** Comparison of RPD values of winter wheat PNC estimation models based on whole and single growth stages.

## 4. Discussion

### 4.1. Estimating Winter Wheat PNC by Parametric Regression (SI)

The estimation of winter wheat PNC based on parameter regression is mainly to build a simple empirical regression model through a narrow hyperspectral band spectral index [9]. The spectral index (SI) is composed of the reflectivity of specific bands, which can partly eliminate the interference of soil, weather, and other factors, and improve the

sensitivity of target parameters. In this study, 10 spectral indexes closely related to crop nitrogen status were used to establish the winter wheat PNC prediction model. Table 3 lists the performance of prediction models for PNC constructed by two spectral indexes with the best fitting degree in the whole and single growth stages. For the whole and each single growth stage, the best performance parametric regression model can explain 64%, 47%, 83%, 86%, and 77% of PNC variability in the booting, heading, flowering, and filling stages. Except for boosting growth stage, other growth period PNC estimation models based on SI are acceptable (RPD > 1.40). It should be noted that the best PNC prediction model for winter wheat in each growth period in Table 2 is based on spectral indexes MTCI and CIre. MTCI and CIre are constructed based on the reflectivity of the red edge position (REP), which indicates the importance of REP in estimating winter wheat PNC. REP is defined as the area where the reflectivity of vegetation increases sharply between 680 nm and 750 nm, which is strongly related to the chlorophyll content [40,64]. Li et al. [65] also showed that compared with other spectral indexes, the spectral index based on REP often had a superior performance when estimating winter wheat leaf nitrogen concentration (LNC). Mutanga et al. [29] found that the correlation coefficient between the spectral index involving the red band and red sideband and the leaf grass nitrogen concentration was the highest when using the field spectral data sampled by the WorldView-2 satellite to predict the leaf grass nitrogen concentration. Similar conclusions have also been found in previous reports [66,67], which is mainly due to the close correlation between nitrogen and chlorophyll content in crops [68–70]. In future research, it will be a continuous trend to estimate crop nitrogen status by building a spectral index based on the hyperspectral remote sensing reflectance of REP.

### 4.2. The Performance of Linear Nonparametric Regressions (SMLR, PLSR)

In this study, two popular linear nonparametric regression models SLMR and PLSR, are selected to predict PNC, and the results are shown in Table 5. The best prediction models for estimating PNC based on linear nonparametric regression methods at booting, heading, flowering, filling, and the whole growth stage of winter wheat are SMLR ($R^2_{testing}$ = 0.64, $RMSE_{testing}$ = 2.69, RPD = 1.58), PLSR ($R^2_{testing}$ = 0.22, $RMSE_{testing}$ = 3.92, RPD = 1.09), PLSR ($R^2_{testing}$ = 0.60, $RMSE_{testing}$ = 1.96, RPD = 1.36), SMLR ($R^2_{testing}$ = 0.83, $RMSE_{testing}$ = 1.17, RPD = 1.80), and PLSR ($R^2_{testing}$ = 0.80, $RMSE_{testing}$ = 1.50, RPD = 1.97). At the booting stage, filling, and the whole growth stage, the prediction model can be used to estimate PNC roughly (RPD > 1.40), and the accuracy of PLSR in the whole growth period is the highest (RPD = 1.97), which almost has strong prediction ability. It shows that PLSR may have more advantages than SMLR when the whole growth stage data are combined. On the one hand, it may be that SMLR always chooses multiple variables at the cost of overfitting and cannot deal with problems such as multicollinearity, resulting in low model accuracy [71,72]. On the other hand, in addition to the positive characteristics screened by SMLR, PLSR also considered the covariance of winter wheat biochemical characteristics at a single growth stage [9]. However, from the perspective of PNC estimation in a single growth stage, the estimation ability of SMLR is not all lower than PLSR, which indicates that SMLR has certain advantages in estimating PNC concentration in a specific growth stage because SMLR has a screening mechanism of spectral index regions of interest [73]. This is only limited to the fact that the relationship between SI and the target variable is well-known and in a specific growth stage. However, the relationship between the explanatory variable and the target variable does not remain unchanged as the growth stage advances. To sum up, the PNC estimation ability of PLSR in the whole growth cc of winter wheat is recognized, but when it comes to each single growth stage, the model changes are diverse, and there is no unified conclusion on which linear nonparametric regression is more advantageous.

### 4.3. The Performance of Machine Learning Regressions (RFR, SVMR, ELMR)

Table 6 lists the performance of prediction models for PNC constructed by RFR, SVMR and ELMR in the whole and single growth stages. The best prediction models for

estimating PNC based on machine learning methods at booting, heading, flowering, filling, and the whole growth stage of winter wheat are SVMR ($R^2_{testing}$ = 0.40, $RMSE_{testing}$ = 3.44, RPD = 1.24), ELMR ($R^2_{testing}$ = 0.91, $RMSE_{testing}$ = 1.42, RPD = 1.87), SVMR ($R^2_{testing}$ = 0.88, $RMSE_{testing}$ = 0.82, RPD = 2.58), RFR ($R^2_{testing}$ = 0.91, $RMSE_{testing}$ = 1.23, RPD = 2.40), and RFR ($R^2_{testing}$ = 0.69, $RMSE_{testing}$ = 2.51, RPD = 1.69). The SVMR model based on the booting stage has the lowest accuracy and does not have PNC prediction ability (RPD < 1.40). The ELMR model based on the heading date and the RFR prediction model of the whole growth stage can be used to roughly estimate PNC, which has a fair estimation ability (1.40 < RPD < 2.00). Both the SVMR model based on the flowering stage and the RFR model based on the filling stage have excellent PNC prediction performance (RPD > 2.00). Zha et al. [52] showed that the accuracy of the RFR model was the best when estimating rice nitrogen nutrition index (NNI) based on UAV remote sensing, and similar conclusions were also found in the research of Reisi Gahrouei [40] and Osco [74]. Wang et al., found that the SVMR model had the highest accuracy when estimating the nitrogen nutrition of tea plants [75]. Wang et al. [8] also found that the SVMR model based on UAV hyperspectral images performed best in estimating nitrogen accumulation in rice leaves. The research showed the potential of machine learning regression SVMR and RFR in the quantitative estimation of crop parameters [38,76–78]. In general, the machine learning methods (RFR and SVMR) constructed in this paper performed best at the flowering and filling stage of winter wheat and had the best estimation performance.

### 4.4. Model Recommendation for PNC

In this study, the winter wheat PNC was quantitatively estimated from UAV hyperspectral image by parametric regression, linear nonparametric regression and machine learning regression, respectively. Except for the booting growth stage, other growth stage PNC estimation models based on parametric regression were acceptable (RPD > 1.40). Linear nonparametric regression (SMLR and PLSR) had no significant improvement in prediction accuracy compared with parametric regression models, no matter in the whole growth stage or single growth stage. This may be because SMLR and PLSR are more suitable for solving some linear regression problems. Relatively speaking, the prediction performance of machine learning regression (RFR and SVMR) is outstanding, especially at the flowering and filling stages (RPD > 2.00). This may be because machine learning regression is more suitable for solving some nonlinear problems, and some more advanced super parameter adjustment methods need to be developed in subsequent research to improve the modeling accuracy of machine learning further [79]. Wang et al. [39] studied the prediction model of rice leaf area index (LAI), and the results showed that RFR and SVMR models could provide more accurate LAI prediction accuracy, which is consistent with the conclusion that SVMR and RFR can provide more accurate prediction results in this study. The estimation of nitrogen status at the later stage of crop growth is of great significance for predicting the yield and quality of wheat grains in the later stage and crops in the next year [80]. The research of Li et al., showed that it was more efficient to estimate the leaf area index at the flowering and filling stage [81]. Fan et al., also showed that the prediction model was more efficient when estimating the nitrogen balance index during the filling stage [67]. To sum up, it was the best choice to estimate PNC at the flowering and filling stage of winter wheat based on hyperspectral reflectance data of UAV imaging by machine learning methods SVMR and RFR, respectively.

### 4.5. Future Works

In this study, winter wheat plant nitrogen concentration (PNC) at different growth stages was quantitatively estimated by combining UAV hyperspectral remote sensing and advanced machine learning algorithms, and encouraging research results were obtained. However, relevant studies have shown that the nitrogen nutrition index (NNI) is a credible indicator for crop nitrogen status assessment, which is of great significance for the proposal of the PNM strategy [21,52,82]. Therefore, the follow-up research work will focus on the

prediction and modeling of winter wheat NNI in the Guanzhong region. At the same time, factors such as climate and management that affect winter wheat growth should be considered as independent variables in nitrogen status assessment so as to build a more stable and reliable guidance for crop precision fertilization. Furthermore, with the gradual maturity of the current satellite imaging and remote sensing technology, the research should be developed from the field scale to the farm scale and a larger region.

## 5. Conclusions

In this study, UAV hyperspectral remote sensing was used to evaluate parametric regression, linear nonparametric regression (SMLR, PLSR), and machine learning regression (RFR, SVMR, and ELMR) to estimate winter wheat PNC at booting, heading, flowering, filling, and the whole growth stages in the Guanzhong area. The results indicated that compared with parametric regression and linear nonparametric regression, the machine learning regression method could obviously improve the estimation accuracy of winter wheat PNC, especially using SVMR and RFR. The calibration set of the model at the flowering and filling stage explained 93% and 92% of the PNC variability, respectively, and the test set of the model at the flowering and filling stage explained 88% and 91% of the PNC variability; $RMSE_{val}$ was 0.82 and 1.23, RPD was 2.58 and 2.40, respectively. Therefore, the conclusion was drawn that it is the best choice to estimate the plant nitrogen concentration at the flowering and filling stage of winter wheat based on hyperspectral reflectance data from UAV imaging. Using machine learning methods, SVMR and RFR, respectively, can achieve the most outstanding estimation performance, which can provide a theoretical basis for putting forward PNM strategies.

**Author Contributions:** Conceptualization, X.C.; methodology, X.C.; software, X.C.; validation, X.C., Q.C. and F.L.; formal analysis, X.C.; investigation, X.C. and B.S.; resources, Q.C.; data curation, X.C. and B.S; writing—original draft preparation, X.C.; writing—review and editing, X.C.; visualization, X.C.; supervision, Q.C., F.L.; project administration, F.L.; funding acquisition, Q.C. All authors have read and agreed to the published version of the manuscript.

**Funding:** This research was funded by the National Natural Science Foundation of China (41701398).

**Data Availability Statement:** Data sharing is not application to this article.

**Conflicts of Interest:** The authors declare no conflict of interest.

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
