# Peer review of "Estimation of Winter Wheat Plant Nitrogen Concentration from UAV Hyperspectral Remote Sensing Combined with Machine Learning Methods"

_remotesensing, doi:10.3390/rs15112831_

Round 1

Reviewer 1 Report (New Reviewer)

Nitrogen and its careful application can significantly improve wheat yields. It is important to know when and in what amount to apply it. The goal of the authors of the present study was to use three different predictive methods to evaluate nitrogen concentration in winter wheat plants (PNC) at the tillering, earing, flowering, filling and throughout the growing season .  For this purpose, statistical and visualization methods were used: unmanned aerial vehicle (UAV) hyperspectral imagery, including parametric regression method; linear nonparametric regression methods-stepwise multiple linear regression (SMLR) and partial least squares regression (PLSR); machine learning methods (random forest regression (RFR), support vector machine regression (SVMR), and extreme learning machine regression (ELMR)), and pay attention to the impact of diferent growth stages on the accuracy of the model. It's a very good idea to combine these methods to estimate nitrogen concentration. The work is very interesting. In the introduction, properly selected literature and well presented topics in similar publications. A big plus in the work is the presentation of photos in the work which allows you to get an even better idea of the subject of the work. The statistical analysis is correctly presented, the tables are clear. Very good presentation of results, which was presented separately for each method. This allows you to better understand the results.  I do not quite understand why the discussion was divided while this is a small minus, because it was very well conducted. Conclusions pertinent and give hope for further development of the topic. 

Author Response

Reviewer 2 Report (New Reviewer)

The authors investigated the estimation of wheat plant nitrogen concentration with UAV remote sensing and developed the inversion model with different methods. My concerns are as follows:

1. In Abstract:  The model for PNC at the flowering and filling stage was developed, how about the model at the booting and heading stage?

2. Introduction: the authors did not summarize the previous results.

3. Lines 109-113: the design is hard to follow. In the N treatment, P and K were not applied. Why in P and K treatments,1/2 N were applied?

4. Figures had low resolution.

Author Response

Reviewer 3 Report (New Reviewer)

The document needs proofreading for language, as there are several errors and unclear sentences throughout. Additionally, there is an excessive use of acronyms which can make it difficult for readers to follow. The objective of the study focuses on nitrogen, yet the experiment also includes treatments of phosphorous and potassium, which could confuse readers. In addition, it is not discussed how this can influence the results.

It is unclear whether ground control points were used, which is an important detail that should be clarified.

Some sections of the methods appear to be disjointed, particularly on page 5. The description of the different regression methods belongs in the introduction chapter as part of the literature review, rather than in the methods section.

The first paragraph of the "Statistical Description of the Winter Wheat PNC" section should be converted into a table to improve readability. The paragraph should then provide an explanation of the results rather than simply describing them.

Figure 4 needs to be improved as it appears blurry and difficult to read. Additionally, the font in the tables is larger than the rest of the text, which can be distracting and inconsistent with the rest of the article.

The Results and Discussion section of the article is challenging to read due to the excessive use of acronyms and statistical results. To improve the clarity of the article, the statistical results could be presented in a table or figure, and the text could focus on analyzing and discussing the results.

By presenting the statistical results in a table or figure, readers can quickly and easily interpret the data without having to navigate through lengthy paragraphs of technical jargon. This would make the article more accessible to a wider audience and would help readers to understand the key findings of the study.

In the text, the focus should be on interpreting the results rather than simply reporting them. The authors should explain what the results mean, how they relate to the objectives of the study, and what implications they have for future research or practical applications.

By presenting the statistical results in a table or figure and analyzing and discussing the results in the text, the article will be more reader-friendly and more likely to be understood and appreciated by a wider audience.

Round 2

Reviewer 2 Report (New Reviewer)

The authors improved the manuscript substantially according to the comments of reviewers. I suggest that the paper can be accepted for publication.

This manuscript is a resubmission of an earlier submission. The following is a list of the peer review reports and author responses from that submission.

Round 1

Reviewer 1 Report

Considering the importance of accurately and effectively obtaining winter wheat nitrogen information to guide the accurate management of nitrogen fertilizer in the field, the author proposed a method for measuring nitrogen concentration in the whole growth period of winter wheat based on UAV hyperspectral images and three statistical methods.

After reading the full text, I found that there are still some deficiencies in the language expression and methods in the manuscript. In order to improve the quality of manuscripts, the following aspects deserve attention:

Line 147&157, please keep the same unit of measurement;

Line 169, why choose these 10 spectral indexes instead of more or less? Please explain;

Line 172-234, this part is a consensus, please simplify the relevant content;

Line 237-253, repetitive expression between text and diagram; It is recommended to delete Table 2 or simplify the statement in Line 237-248; The proposal is applicable to the full text;

Line 257, please draw the correlation heat diagram between spectral index and PNC to confirm the best fit of MTCI and CIre mentioned by the author;

Other aspects:

In the third section, although the modeling accuracy differences of the three PNC prediction methods in the whole growth period and each growth period are discussed, I am most confused that the author directly established the PNC prediction model by using the unselected spectral index; In general, there may be a high degree of autocorrelation between too many features. As feature dimensions increase, simulation performance may decline. It is suggested that the author should use some parameter sensitivity analysis methods to flexibly select an appropriate number of characteristic spectral indexes to build PNC prediction models.

Reviewer 2 Report

The study Estimation of Winter Wheat Plants Nitrogen Concentration Using Unmanned Aerial Vehicle Hyperspectral Remote Sensing Combined with Machine Learning Methods is very interesting.

 I as a lover of remote sensing/UAV/machine learning liked this work very much. However, some points concern me, mainly regarding some consistency in the results that I believe the authors should clarify. 

The authors used a sensor with 125 channels, however to model they used, at most, 10 predictors (Table 1). I found this to be a very weak point. The authors could use the spectral bands separately and/or together with indices. They could also perform an inter-band normalization, in which a normalization would be generated among the 125 sensor bands and then use filter elimination to select the best predictors. See these two papers that used inter-band normalization and filter elimination (https://doi.org/10.3390/atmos13091518 e https://doi.org/10.1016/j.eja.2019.03.001)

The impression I have is that the authors used a Ferrari at a speed of 40 km/h.

Abstract: replace the word validation with test. The proper term for checking the performance of a model is called testing. Do this for all text

Keywords: replace the words already in the title

2. Materials and Methods:

In line 132 and 133 the resolution of 1000 x 1000 pixels and 50 x 50 pixels is mentioned. Is this the spatial resolution? If so, what is the unit? m, cm etc.

The methodology does not explain how the separation of the data set for training and the data set for testing was performed.

It also does not explain how the spectral indices were used. Were they used on a specific date or were they integrated or accumulated until the date of interest?

3. Results

In Table 3 the Growth stages Booting validation for MTCI has an R² of 0.47 equal to r of 0.68. Figure 4 for this model shows a large scatter, which visually appears to have no correlation. Please check this data.  

Table 5 and Figure 5 also shows the same inconsistency. As you can see, the Filling PLSR has an R² of 0.80 and in Figure 5 you see that there is absolutely no correlation.

Also in Table 5, it is observed that the SMLR and PLSR models for some Growth stages had overfitting. 

It would be interesting if the authors extracted the most important variables. Using the varImp function from the Caret package for R this is possible. Thus, they could analyze which variable had the most weight in the models.

Reviewer 3 Report

I’d like to compliment the authors' ideas on UAV hyperspectral analysis for nitrogen content estimation of winter wheat through various growth stages. However, when reading the manuscript, I had some trouble figuring out the novelty of the research. Given that the proposed machine learning approaches are commonly used, many studies have adopted them for hyperspectral analysis. The authors collected only 36 samples for each growth stage and the nitrogen contents demonstrate very large standard deviation values, so I don’t think the developed models are reliable. I would like to reject this manuscript. Some suggestions and comments are provided below to improve the work.

Major comments

1. The introduction section should be reorganized to emphasize scientific questions and the authors’ contribution.

2. The authors have compared the prediction performance of different models, and I suggest to interpret why the machine learning model obtains high accuracies.
